# An Improved 3D Deep Learning-Based Segmentation of Left Ventricular Myocardial Diseases from Delayed-Enhancement MRI with Inclusion and Classification Prior Information U-Net (ICPIU-Net)

**DOI:** 10.3390/s22062084

**Published:** 2022-03-08

**Authors:** Khawla Brahim, Tewodros Weldebirhan Arega, Arnaud Boucher, Stephanie Bricq, Anis Sakly, Fabrice Meriaudeau

**Affiliations:** 1ImViA EA 7535 Laboratory, University of Burgundy, 21078 Dijon, France; khawla.brahim@u-bourgogne.fr (K.B.); tewdrosw@gmail.com (T.W.A.); arnaud.boucher@u-bourgogne.fr (A.B.); stephanie.bricq@u-bourgogne.fr (S.B.); 2National Engineering School of Sousse, University of Sousse, Sousse 4054, Tunisia; 3LASEE Laboratory, National Engineering School of Monastir, University of Monastir, Monastir 5000, Tunisia; sakly_anis@yahoo.fr

**Keywords:** segmentation, ICPIU-Net, myocardium, myocardial infarction (MI), late gadolinium enhancement magnetic resonance (LGE-MR), microvascular-obstructed (MVO), deep learning

## Abstract

Accurate segmentation of the myocardial scar may supply relevant advancements in predicting and controlling deadly ventricular arrhythmias in subjects with cardiovascular disease. In this paper, we propose the architecture of inclusion and classification of prior information U-Net (ICPIU-Net) to efficiently segment the left ventricle (LV) myocardium, myocardial infarction (MI), and microvascular-obstructed (MVO) tissues from late gadolinium enhancement magnetic resonance (LGE-MR) images. Our approach was developed using two subnets cascaded to first segment the LV cavity and myocardium. Then, we used inclusion and classification constraint networks to improve the resulting segmentation of the diseased regions within the pre-segmented LV myocardium. This network incorporates the inclusion and classification information of the LGE-MRI to maintain topological constraints of pathological areas. In the testing stage, the outputs of each segmentation network obtained with specific estimated parameters from training were fused using the majority voting technique for the final label prediction of each voxel in the LGE-MR image. The proposed method was validated by comparing its results to manual drawings by experts from 50 LGE-MR images. Importantly, compared to various deep learning-based methods participating in the EMIDEC challenge, the results of our approach have a more significant agreement with manual contouring in segmenting myocardial diseases.

## 1. Introduction

Cardiovascular diseases (CVDs) are the leading cause of global mortality, with an estimated 17.9 million deaths in 2019, mainly due to MI. Radiologically diagnosing MI in its early phases plays a crucial role in supplying guidance on further patient treatment. In recent decades, automatic methods have been developed to improve the diagnosis and prognosis steps of CVDs. A crucial clinical parameter to evaluate the state of the heart after MI is the viability of the considered segment, i.e., if the segment recovers its functionality upon revascularization.

Replacement myocardial fibrosis is a known substrate for occurring malignant ventricular arrhythmias (VA), a prevalent cause of abrupt cardiac death worldwide. The scar development in the heart is most commonly from MI, an irreversible decease of the contractile muscle associated with the occlusion of a coronary artery. MI occurs as a result of atherosclerosis, in which plaque builds up inside the artery walls. This build-up makes the arteries progressively narrower and slows blood flow, causing angina. Finally, an area of cholesterol plaque can tear inside of a coronary artery. This rupture results in a blood clot forming on the plaque’s surface, which can then completely block blood flow through arteries. If the blockage is not remedied fast, the heart muscle begins to die. The healthy heart area is substituted with the infarct area. In chronic MI, capillaries in the myocardial region continue to be impeded after the re-perfusion, indicating severe ischemic disease. MVO, also called the no-reflow phenomenon, is an incident that usually appears in a proportion of subjects with acute MI following re-perfusion therapy of an occluded coronary artery [1]. Patients sustaining MVO regions have higher proportions of MI and raised mortality. Recently, computational modeling for the accurate characterization of myocardial scar geometry and its volume and the heterogeneity of patients with chronic ischemic cardiomyopathy (IC) may help clinicians determine the appropriateness of analysis and treatment-related rhythm disorders [2,3].

Two-dimensional (2D) LGE-MRI is the primary reference for recognizing myocardial scarring through enhancement from preserving its based contrast agents. LGE-MRI is a non-invasive technique achieved approximately after 10 min of gadolinium-based contrast agent injection. Healthy and infarct tissues are distinguished by their altered wash-in and wash-out contrast agents. Nonetheless, by progress in MRI acquisition techniques, 3D LGE-MRI has arisen with improved spatial resolution, allowing accurate volumetric quantification of scar tissue [4,5,6,7]. As the manual segmentation is tedious, dependent on observer variability, and time-consuming, automated volume segmentations are highly intended for this task. This increased interest was primarily justified through the success rate performed by these methods. There are several studies on automatic MI segmentation methods that motivate our approach.

Considering the performance reached by deep-learning approaches in medical image analysis, this paper proposes an inclusion and classification of prior information U-Net-based network (ICPIU-Net) for the fully automatic and efficient segmentation of LV healthy myocardium and LV myocardial diseases in LGE-MRI. Our approach integrates image features and a post-processing decision phase to aggregate the final prediction. This algorithm is novel to quantify myocardial tissues’ presence on a set of contrast-enhanced acquisitions, leading to better prevention and higher survival opportunities for patients.

Figure 1 shows short-axis LGE-MRI at the base, middle, and apical ventricular levels, illustrating a large region of hyperenhancement (scar) with a hypo-enhanced central area (MVO).

The rest of the paper is structured as follows: In Section 2, we introduce the previous literature related to our study. In Section 3, we describes the new methodology used throughout this work. In Section 4, we present experimental results that allow the quantitative evaluation of the performance of our approach on the EMIDEC test dataset and a comparison between different methods. Finally, in Section 5, we summarize the main conclusions of the paper.

## 2. Related Work

Traditional scar segmentation research studies were frequently based on intensity thresholding or clustering techniques responsive to local intensity variations [8]. The predominant limitation of these techniques is that they typically require expert delineations of the region of interest to decrease the computational costs and the search space [9]. As a result, researchers focus on deep-learning cardiac segmentation approaches that are more convenient for clinical guidance. For instance, Moccia et al. [10] applied fully convolutional networks (FCNs) for MI-tissue segmentation protocols from LGE-MRI. The authors investigated two segmentation algorithms. Segmentation results against expert contours showed that both algorithms identified scar tissues in LGE-MRI, particularly when delimiting the search area to the myocardium only. Xu et al. [11] proposed a new joint motion feature-learning architecture based on deep learning and optical flow to segment MI from non-contrast agents cardiac MRI accurately. The validation results proved that the suggested architecture has a comparable performance with a human expert’s delineation (pixel-level accuracy: 95.03%, Kappa statistic: 0.91, Dice: 89.87%, and Hausdorff distance: 5.91 mm). De La Rosa et al. [12] proposed a deep learning-based method for the automatic segmentation and quantification of the scar and MVO tissues in LGE-MRI. Their approach is based on a cascade framework where, firstly, healthy and diseased slices are distinguished by a convolutional neural network. Secondly, the MI is segmented by an initial fast coarse segmentation. Then, the resulting segmentation is refined by a boundary-voxel reclassification strategy to incorporate MVO tissues in the infarction segmentation. Compared to the reference techniques, the proposed network achieved the highest agreement in volumetric infarct segmentation with the manual delineations (*p* < 0.001). This method reached an average Dice coefficient of 77.22 ± 14.3% and a volumetric error of 1.0 ± 6.9 cm. Hao et al. [13] developed a multi-branch fusion architecture for automatic MI screening from 12-lead ECG images. Their method included feature fusion and classification network. Extensive experiments demonstrated that the proposed architecture reached human-level performance on all four evaluation criteria (accuracy, sensitivity, specificity, and F1-score of 94.73%, 96.41%, 95.94%, and 93.79%, respectively).

U-Net [14] has become the widespread variant of FCNs for biomedical image segmentation and is commonly employed in cardiology. The network uses skip connections between the down-sampling and up-sampling paths to recover the spatial context loss in the encoder, performing more accurate segmentation. Different previous cardiac image-segmentation networks have utilized the 3D U-Net [15] and the 3D V-Net [16] as their basis architectures, reaching efficient segmentation for several cardiac segmentation tasks [17,18,19]. Isensee et al. [20] introduced nnU-Net (no-new-U-Net) to automatically adjust preprocessing techniques and network architectures to a medical dataset. nnU-Net has been well applied to several segmentation tasks [21,22]. Despite the potential of deep learning for several fields, few deep learning-based methodologies have been proposed in the literature for infarct segmentation from LGE-MRI. Fahmy et al. [23] used a U-Net architecture with 150 layers to automatically quantify LV mass and infarct volume on LGE images of 1041 subjects with hypertrophic cardiomyopathy (HCM). Their methodology reported DSCs of 82 ± 0.08% (per-patient) and 81 ± 0.11% (per-slice) for the LV quantification and 57 ± 0.23% (per-patient) and 58 ± 0.28% (per-slice) for MI segmentation. In another study, Zabihollahy et al. [24] developed a U-Net-based network to accurately segment LV myocardium and infarct borders from 3D LGE-CMR images of 34 subjects with IC. Two cascaded subnets were used to segment the LV myocardium and quantify the MI region into the segmented LV myocardium. Three U-Nets were trained in each subnet using slices extracted from coronal, axial, and sagittal planes. The proposed network reached a *DSC* of 88.61 ± 2.54% for LV infarct segmentation on the 3D test dataset. Recently, Arega et al. [25] proposed a segmentation network that, firstly, generates uncertainty estimates during its learning process using the Monte Carlo dropout method. Secondly, it integrates the uncertainty information into the loss function for better segmentation results. The proposed model showed an accurate segmentation of all myocardial regions.

The major disadvantage of adopting 2D and 3D FCNs is that they are usually trained with cross-entropy, soft-Dice losses, and compound loss functions. These losses neglect high-level features that represent the implicit anatomical structures, such as their shape or topology and spatial relationships between tissues [26]. Likewise, U-Net architecture does not influence contextual or anatomical consistencies. Several research works focus on alleviating this challenge by incorporating further prior information to improve network robustness and produce plausible segmentations [27]. Prior knowledge can be introduced into segmentation in many ways, such as by appending it as a penalty term in the loss function, anatomical, or contextual constraints. Many researchers have used autoencoders to extract semantic feature information from input images or labels, which steer the cardiac image segmentation [28,29,30]. The contextual information can include shape priors to guide the segmentation results toward a ground truth shape. Oktay et al. [31] modified the decoder layers of a U-Net architecture to embed prior information through super-resolution gold standard maps using cardiac cine MRI. Zotti et al. [32] developed a grid-Net-based network to segment heart structures from cardiac cine-MRI. Their model integrates cardiac shape prior information to encode a 3D position-point likelihood for being a definite class. More recently, El Jurdi et al. [33] included position and shape priors to the learning phase via inserting bounding filters on the skip-connections in a U-Net model. Duan et al. [34] introduced a shape-constrained bi-ventricular segmentation technique. Firstly, a multi-task network is used to localize definite landmarks. Then, these landmarks are employed to initialize atlas propagation during the segmentation step to improve the segmentation quality. These networks can also be adjusted for improving spatial, temporal, and topology consistency of segmentation prediction in the cardiac cycle [35,36,37,38,39,40].

Different studies based on deep learning models evaluating infarct segmentation from LGE-MRI have been included in the EMIDEC challenge (http://emidec.com/) (accessed on 1 April 2020). Camarasa et al. [41] presented two approaches to evaluate if the uncertainty of an auxiliary unsupervised task is helpful for MI segmentation. Their baseline method first determined the ROI centered on the non-background labels to use U-Net architecture to segment all myocardial regions from the definite ROI. Feng et al. [42] developed an automatic LGE-MRI segmentation model using: (a) rotation-based augmentation to force the algorithm to remove the image orientation and learn the anatomical and contrast relationships; (b) dilated 2D U-Net to increase the robustness of the network against different slices’ misalignment. The authors applied the weighted cross-entropy and soft-Dice loss functions to relieve the class imbalance problem. They also favored slices containing damaged tissues. Girum et al. [43] proposed a two-stage CNN network to segment the anatomical structures firstly, and then pathological regions from LGE-MRI. The segmented myocardium area from the anatomical network is further used to refine the pathological network’s segmentation, thus producing the final four-class segmentation output. Huellebrand et al. [44] compared a hybrid mixture model approach with two U-Net segmentations. The proposed mixture model is inspired by [45] and is suited to EMIDEC data. This algorithm differentiated the infarct regions depending on the intensity distribution. The authors proved that a better segmentation is achieved using a mixture of Rayleigh and Gaussian than a mixture of Rician and Gaussian. In addition, they realigned the image slices to prevent any inconvenience due to respiratory motions. Yang and Wang [46] developed an improved and hybrid U-Net architecture for myocardial segmentation in LGE-MRI. The modified U-Net embodied the squeeze-and-excitation residual (SE-Res) module in the encoder part and a selective kernel (SK) block in the decoder part. Zhang [47] proposed a cascaded convolutional neural network to segment myocardial zones from LGE-MRI automatically. Its model achieved the best segmentation performance. The winner first employed 2D U-Net to focus on the intra-slice information for a preliminary segmentation and then a 3D U-Net to focus on the volumetric spatial information for a better segmentation based on both the original volume and the 2D segmentation. Finally, post-processing, removing all the scattered pixels from the latest segmentation, is applied to produce the final segmentation. Zhou et al. [48] developed an anatomy prior-based network, which combines the U-Net segmentation architecture with attention blocks. They also presented a neighborhood penalty strategy to assess the inclusion relationship among the healthy myocardium and damaged areas, and a data-augmentation technique based on the mix-up strategy [49] to interpolate two images and their corresponding segmentation maps.

## 3. Materials and Methods

### 3.1. Study Subjects and Data Acquisition

The LV myocardial diseases LGE-MRI dataset used in our experiment is provided by the EMIDEC challenge [50]. The data was acquired with 1.5 and 3 T Siemens MRI scanners. Additional data (12 features), including physiological and patient medical background were also included but not used in the present study. The training dataset includes 100 subjects, among them 67 scar cases and 33 normal cases. The EMIDEC test dataset consists of 50 exams divided into 1/3 healthy cases and 2/3 unhealthy myocardial cases (Table 1). All subjects underwent a standardized imaging protocol of LGE-MRI. Each patient’s LGE-MRI contains a series of 5–10 short-axis slices, wrapping the whole LV myocardium from the base to the apex. Manual boundaries (LV cavity, LV healthy myocardium, scar, and MVO), delineated by a biophysicist with deep competency in medical imaging, are presented for the training set.

### 3.2. Pre-Processing

The data pre-processing is one part of the pipeline. We performed pre-processing steps to the test data similar to those that were performed on training data. It includes the following tasks.

To reduce the class imbalance in medical images and eliminate unimportant anatomical structures, all images were cropped automatically to the region whose center is the centroid of the LV cavity. LGE-MRI cropping was fully automatic. Cropping results in both reducing the background class as well as the computational time for training our model.

Since the EMIDEC data shape varies between various subjects, it is necessary to alleviate the shape mismatch by reshaping each patient’s volume image. Thus, all MR volumes are reshaped to 96×96×16 with appending empty slices [51].

A standard adaptive histogram equalization algorithm is then used to enhance the local image contrast [52,53] and, hence, improving the efficiency of the training process. We also applied the non-local mean denoising [54] to all the data for decreasing the noise.

### 3.3. Architecture of ICPIU-Net

Figure 2 presents the pipeline of the ICPIU-Net network, which includes two major stages (Anatomical-Net (nnU-Net [20]) and Pathology-Net) as the myocardial diseases (MI and MVO) that are ensured to be localized within the LV myocardium. By segmenting the LV myocardium and cavity first, we may eliminate other hyper-enhanced and hypo-enhanced tissues of the LGE-MRI.

A schematic flowchart of the ICPIU-Net network is displayed in Figure 3. In the training step, Anatomical-Net and Pathology-Net were trained separately on 100 LGE-MRI. In the testing step, 50 LGE test MR images were supplied to the trained algorithm to generate the relative segmentation maps, which were later fused through a majority voting technique to yield the final label prediction of myocardial segmentation output. Each stage’s details are explained in the following clauses.

#### 3.3.1. Anatomical Network

The segmentation networks used in the anatomical network are based on nnU-Net [20]. nnU-Net is a fully automatic segmentation framework based on the widely used U-Net [14] architecture. Similar to U-Net, it uses convolutional layers between poolings in the down-sampling path and deconvolution operations in the up-sampling path. However, it replaces classical rectified linear unit (ReLu) activation functions with leaky ReLus (leakiness = $10^{-2}$) and uses instance normalization [55] rather than the more common batch normalization (BN) [56]. The order of operations is as follows: convolution—instance norm—leaky ReLus. In addition, the downsampling is completed using strided convolutions instead of max-pooling. nnU-Net ensembles 2D U-Net and 3D U-Net networks. 2D U-Net trains whole slices to concentrate on intra-slice information. A 3D U-Net is used to extract the volumetric spatial features. This architecture aims to restrict the impact of intra-slice heterogeneity and take into account the volumetric information for more accurate segmentation. Thus, the cross-validation outputs automatically lead to the best ensemble to be employed for the testing prediction.

The anatomical network uses 2D, 3D, and cascaded U-Net to alleviate the convenient shortcoming of a 3D U-Net architecture on datasets with large-size images. In a cascaded U-Net, a 3D U-Net is first trained on 3D down-sampled images for a preliminary segmentation. Then, the resulting segmentations are up-sampled to the original input resolution and passed to a second 3D U-Net, trained on patches at full resolution for final prediction segmentation.

The anatomical network was implemented on NVIDIA Tesla V100 with four embedded GPUs using the Pytorch source code (http://github.com/MIC-DKFZ/nnunet/) (accessed on 1 February 2021) based on the nnU-Net implementation [20]. To train the model, we employed a five-fold cross-validation. The ADAM optimizer was used with an initial learning rate of $3\times 10^{-4}$. The learning rate is reduced during training using a polynomial learning rate scheduler. The short-axis slice and volume inputs are provided for 2D and 3D networks, respectively. The sum of the cross-entropy loss ($L_{CE}$) and the Dice loss ($L_{DICE}$) is used as the final loss function to train the proposed network (Equation (Equation 1)):(1)$$L=L_{CE}+L_{DICE}.$$

The $L_{DICE}$ function is summarized as follows:(2)$$L_{DICE}=-\frac{2}{\left|K\right|}\sum_{k\in K}\frac{\sum_{i\in I}u_{i}^{k}v_{i}^{k}}{\sum_{i\in I}u_{i}^{k}+\sum_{i\in I}v_{i}^{k}},$$
where *u* refers to the softmax output of the proposed network and *v* is a one-hot encoding of the ground truth label delineated manually by the experts. Both *u* and *v* have shape I×K with i∈I being the pixels’ number in the training patch/batch and k∈K being the different classes.

#### 3.3.2. Pathological Network

##### 3D U-Net Architecture

Our pathological network mainly has a 3D U-Net as the main network and it incorporates some prior knowledge to further improve the result. The encoder part consists of stacked convolutions followed by BN, ReLu, and max-pooling layers for capturing contextual information. The decoder part consists of deconvolutions, convolutions, BN, and ReLu for the accurate position of patterns. Skip connections concatenate symmetrically contextual and positional attributes from opposing contracting and expanding pathways. The last convolution layer reduces its output channels number to the number of predicted classes, generating a myocardial segmentation map of identical dimensions to that of the target map.

##### Network Implementation

We trained the proposed architecture with sampled patches of size 12×12×12 voxels and a batch size of 4. The training is completed using the ADAM optimizer with a learning rate $\alpha =10^{-4}$ for a maximum of 200,000 iterations, taking a total time of 314 min. The pathological network was implemented in Python using the Chainer library.

##### Shape Reconstruction

To learn the latent representation from which the original cardiac shapes can be reconstructed with inclusion-restricted segmentation, we train the proposed 3D convolutional variational autoencoder (CVAE), which uses an iterative optimization process with expert ground truth. The 3D CVAE encodes the original cardiac shape information as a compact representation in a reduced dimension, interprets the code, and decompresses the input without any reconstruction loss. Thus, a pre-trained 3D CVAE from an ensemble of cardiac shapes is used as a constraint to regularize a segmentation output into a proper shape [30]. The 3D pre-trained CVAE has in-depth information about segmenting several feature representations’ categories. Compared to [30], our CVAE learns the general shape as well as the inclusion of the MVO into the MI itself into the myocardium. The inclusion criteria are helpful for plausible reconstruction with accurately localizing the borders of the cardiac tissues. Figure 4 depicts the configuration of the proposed 3D CVAE.

##### Class Constraint

All along the optimization process, we develop a binary classification module to distinguish patients with myocardial infarction from regular patients. Hence, we incorporate classification priors for the segmentation process to constrain the predicted tissue in this known category.

As Figure 4 illustrates, we propose the inclusion (*IC* in cyan) and class constraint (*CC* in purple) modules, connected as an extended network and to the bottom of the 3D U-Net, respectively, for penalizing the final prediction of myocardial segmentation output. These constraints are introduced as auxiliary $L_{IC}$ and $L_{CC}$ loss terms to highlight small classes’ tissue.

One of the main challenges with diseased myocardial tissues’ segmentation is the class imbalance (i.e., LV healthy myocardium has way more instances than pathological regions) in the dataset. As shown in Table 1, the EMIDEC test dataset provides 1/3 healthy patients and 2/3 infarcted patients. The resulting segmentation of training with the cross-entropy loss function may not be effective, as the most frequent class may leverage training. That is why it is critical to optimize the appropriate loss function for accurate segmentation. We train our penalty-based pathological network with a fusion of multi-class mean intersection over union (IoU) loss $L_{Seg}$ [57], inclusion constraint loss $L_{IC}$, and a class constraint loss $L_{CC}$. This final loss function is given in Equation (Equation 3):(3)$$L_{Final}=L_{Seg}+\lambda_{IC}\times L_{IC}+\lambda_{CC}\times L_{CC},$$
where $\lambda_{IC}$ denotes the inclusion constraint penalty-term and $L_{IC}$ indicates the L2 loss function that is defined in Frobenius norm Equation (Equation 4); $\lambda_{CC}$ represents the class constraint penalty-term and $L_{CC}$ indicates the cross-entropy loss function. We regularize, with $\lambda_{IC}$ and $\lambda_{CC}$, the weights in the interval ($10^{-2}$, $10^{-1}$).
(4)$$L_{IC}=\sum_{\begin{array}{c} i=1 \end{array}}^{n}{\left|\left|RP_{i}-RG_{i}\right|\right|}_{F}^{2},$$
where *n* is the total number of training volumes, $RG_{i}$ is the reconstructed manual delineation, $RP_{i}$ is the reconstructed segmentation output, and ${\left|\left|.\right|\right|}_{F}$ denotes the Frobenius norm of an m×n matrix.

The multiclass $L_{Seg}$ function measures the overlap between two samples [58] and is incorporated into deep learning networks as follows:(5)$$L_{Seg}=L_{IoU}=\frac{1}{\left|C\right|}\sum_{\begin{array}{c} c \end{array}\in C}\frac{\sum_{\begin{array}{c} i \end{array}}p_{ic}\times p_{ic}^{*}}{\sum_{\begin{array}{c} i \end{array}}p_{ic}+p_{ic}^{*}-p_{ic}\times p_{ic}^{*}},$$
where $p_{ic}$ is the prediction score at position *i* for class *c*, and $p_{ic}^{*}$ is the gold standard distribution which is a delta function at $yi^{*}$, the true label.

### 3.4. Post-Processing

In our post-processing, we employed morphological operators such as opening (kernel size of 3×3), to remove small predicted classes with less than 64 voxels from the predicted segmentation. In addition, we used connected components to further improve the segmentation result of scar and MVO. Finally, the cropped slices were resized to the original input LGE-MRI size.

The majority voting method, based on all fusions of our models’ results, with varying estimated parameters from training ($\lambda_{IC}$ and $\lambda_{CC}$), is used to improve the segmentation. Based on the best experimental results, we chose $\lambda_{IC}$∈$\left[10^{-2},5\times 10^{-1}\right]$ with an increment step of $7\times 10^{-2}$ and $\lambda_{CC}$∈$\left[10^{-2},5\times 10^{-1}\right]$ with an increment step of $7\times 10^{-2}$, obtaining the best fit. Indeed, hyperparameter tuning ranges from $10^{-2}$ to $5\times 10^{-1}$ make for the best trade-off between evaluation metrics. Thus, the voxel was labeled as infarct if at least three of the combined models predicted this voxel as an infarct label. The final model (or ensemble) that yields the highest Dice similarity coefficient (*DSC*) on the training set is automatically selected.

## 4. Results and Discussion

### 4.1. Evaluation Metrics

As proposed by the challenge organizers, we employed region-based and volume-based evaluation metrics (in 3D) to appraise the performance of our approach-generated segmentations of myocardial tissues. The *DSC* (Equation (Equation 6)) computes the spatial overlap of our presented model delineation and the gold standard, varying from 0 (mismatch) to 1 (excellent match). A various class of scores evaluates the distance between segmentation contouring.
(6)$$DSC=2\frac{\left|P\cap G\right|}{\left|P\right|+\left|G\right|},$$
where *P* and *G* represent the predicted and manual segmentation maps, respectively.

Given two boundaries generated from our algorithm ($A=\left\{a_{i}:i=1,\cdots ,N_{A}\right\}$) and manual ($M=\left\{m_{j}:j=1,\cdots ,N_{M}\right\}$) segmentation, the Hausdorff distance (*HD*) [59] is determined as follows:(7)$$HD=\max_{a_{i}\in A}\min_{m_{j}\in M}\left|\left|a_{i}-m_{j}\right|\right|.$$

The *HD* computes the degree of mismatch between *A* and *M* by calculating the Euclidean distance of each point $a_{i}$ that is most distant from any point $m_{j}$.

The absolute volume difference (*AVD*) calculates the difference between our method-generated $V_{A}$ and manual $V_{M}$ LV volumes by the expert. In addition to the *AVD*, absolute volume difference rate according to the volume of the myocardium (*AVDR*) metric, Equation (Equation 8) was reported:(8)$$AVDR=\frac{AVD}{V_{MYO}},$$
where $AVD=\left|V_{A}-V_{M}\right|$ and $V_{MYO}$ is the myocardium volume of the manual annotation.

For consistency with other publications, the metrics were based on the online evaluation platform (http://github.com/EMIDEC-Challenge/Evaluation-metrics/) (accessed on 1 April 2020). Therefore, region- and volume-based metrics were measured for each test patient, and we calculated their mean values to investigate the performance of our approach for myocardial disease segmentation. We also applied Bland–Altman plots [60] to consider the accuracy between the proposed method and manually generated LV volumes.

### 4.2. Results Analysis and Extensive Discussions

We compared the results of our proposed network to different previous methods used in the EMIDEC challenge, enclosing Feng et al. [42], Huellebrand et al. [44], Yang et al. [46], Zhang [47], Camarasa et al. [41], Zhou et al. [48], and Girum et al. [43]. The LV myocardium, scar, and MVO were segmented using those methods from the same test dataset. We also compared the ICPIU-Net segmentation results to Brahim et al. [61], which is based on only a shape prior constraint for myocardial disease segmentation and manual delineations. The results prove that our approach performs well on all substructures. A statistical test was conducted for each model to check if the difference of the results between the coronary arteries is significant. We have found that there is no statistical difference.

There are a total of 100 exams with published labels to train our algorithm. We make random five-fold cross-validations by shuffling the scan sequence and splitting the database into five folds to provide a more comprehensive evaluation of our network. In Table 2 and Table 3, the cross-validation results of our segmentation output and of two networks’ segmentation output are represented.

The conducted experiments’ metrics on the internal cross-validation shown in Table 2 prove that our approach can accurately segment each target tissue despite using a small sample size. The standard deviations, which are relatively small, demonstrate the stability of our proposed method for the segmentation of myocardial diseases.

Table 3 gives a summary of the comparison study. From the result, our approach yielded the best clinical and geometrical metrics compared with two existing networks.

Figure 5 shows the LV myocardium, infarct, and MVO segmentation results of Brahim et al. [61] and our proposed ICPIU-Net for three different slices, randomly chosen from three patients of the testing dataset. We stacked up the segmented 2D slices of each EMIDEC test dataset to reproduce a 3D rendering surface of myocardial regions for visualization purposes. In comparison to Brahim et al. [61], our approach segmented myocardial diseases more accurately. Visually, the proposed network-generated segmentation narrowly matches the manually segmented delineation for all the labeled regions. The test results proved that our approach is comparatively accurate in segmenting scar tissue.

Figure 6 shows segmentation results of EMIDEC challengers, ground truth mask, and our proposed ICPIU-Net for all slices, chosen from one patient of the testing dataset. Visually, our proposed network correctly depicted myocardial structures and showed a good agreement with the gold standard. In comparison to Zhang’s method, our approach demonstrated promising performance in segmenting the damaged myocardial areas from LGE-MRI.

As shown in Figure 7, qualitative evaluations demonstrate that our proposed network accurately segments infarcted patients, especially at the middle slices. The segmentation results achieved by our proposed ICPIU-Net approach are coincident with the expert delineations for both two volumes. Most segmentation errors appear at basal and apical short-axis slices.

Table 4 summarizes the quantitative results of our proposed method against those of the alternatives for the testing dataset. Experimental results demonstrate that the myocardium segmentation is globally satisfying, whereas the diseased areas (i.e., MI and MVO) are comparatively challenging to predict accurately. The evaluation methods were based on the clinical metrics most applied in cardiac clinical practice: the *AVD* and *AVDR* of the diseased areas (MI and MVO) and the geometric metrics: the *DSC* for the different tissues and *HD* for the myocardium. A ranking is computed for each metric, and the final ranking representing the sum of the rankings for each evaluation metric is chosen to select the best model. Our developed ICPIU-Net reported higher *DSC*, *AVD*, and *AVDR* than other methods for MI and MVO segmentations. The second-best *DSC* for fully automated segmentation of myocardial diseases was reached using the network proposed by Zhang [47] (71.24% for scar and 78.51% for MVO). In testing, our proposed method also achieved *DSC*, *AVD*, and *HD* of 87.65 %, 8863.41 mm${}^{3}$, and 13.10 mm for LV myocardium segmentation, respectively. The publicly available test database consists of 50 exams divided into 17 cases with normal MRI after injection of a contrast agent and 33 patients with myocardial infarction. A patient-by-patient study of the proposed network-generated segmentation revealed that the infarct tissue could be correctly determined in 32 out of 33 pathological subjects from the test dataset.

We conducted a comprehensive ablation study of prior information modules to investigate their impact on the segmentation results. As shown in Table 5, the combination of *IC* and *CC* regularization penalty terms provided more plausible segmentation close to the manual delineation. The results have demonstrated the effectiveness of inclusion and class constraints to the segmentation task on the EMIDEC dataset.

The metrics of *DSC*, *AVD*, and *AVDR* applied during the MVO segmentation challenge can be inconsistent since the MVO represents only a small volume of the input LGE-MRI. Indeed, MVO’s absence from all the data seemingly supplies correct results with *DSC* and volumes. However, the accuracy highlights the effectiveness of the proposed method to identify MVO regions. Therefore, in Table 6, the additional metrics of MVO tissue are provided. Results reveal the pertinence of inclusion and class constraints in segmenting MVO tissues.

Figure 8 depicts the Bland–Altman plots for the proposed ICPIU-Net vs. expert manual LV volumes. In these graphics, the dashed blue line, and the upper and lower red dashed lines show the mean value of the difference and the upper and lower limits of accordance, respectively. In comparison to manually segmented volumes, our method’s mean bias in assessing LV myocardium, MI, and MVO volumes was 4.9888 cm${}^{3}$, 1.2266 cm${}^{3}$, and 0.5112 cm${}^{3}$, respectively. Thus, our proposed network lightly overvalued the LV myocardial tissue volumes, resulting in a mean absolute LV volume percentage error of 8.12%.

The EMIDEC classification contest aims to classify the patients as healthy or infarct. In Table 7, the classification results of each method are provided. The proposed network achieved the best results with the best pathology classification accuracy of 98%. Our approach failed only on 1 exam among 50, which the challenge organizers considered to be an outstanding result.

Deep learning networks have significantly boosted state-of-the-art segmentation performance in cardiac MRI. Evaluation of the MI’s presence and extent (MVO) stays essential for assessing myocardial viability. The visual estimation by cardiologists remains the routine approach. However, an accurate automatic prediction of the exams as an objective assessment of the volume and the percentage of damaged myocardium plays a crucial role in treating and improving clinical outcomes. The proposed network achieved a Dice score of 0.8765 for the myocardium and 0.7336 for the infarction tissue. Nevertheless, MI tissue segmentation still proved to be a challenging task compared to the myocardium. Our results demonstrate that automatic myocardial segmentation is now a possible task. Still, the segmentation of diseased regions requires further development before being included in software solutions applied in clinical practice.

## 5. Conclusions

In this paper, we describe a novel deep learning-based approach for fully automated segmentation of myocardial tissue in LV myocardium from LGE-MRI. The experimental results proved the relevance of our proposed architecture for clinical guidelines diagnosis and treatment planning. Our ICPIU-Net outperformed prior deep learning-based techniques in terms of segmentation accuracy. In building our approach, we addressed the critical class imbalance issue due to a relatively small size of myocardial diseases compared to the healthy area in the myocardium via constructing informative inclusion and classification constraints from pathological tissues. A decision-fusion technique was used to aggregate the predictions achieved through varying estimated parameters from training for final prediction. Nonetheless, cases of MVO were not all identified in the subject cohort enlisted for this work (Acc. of 84.00%). Thus, further studies, to integrate the clinical metadata information, are needed for improving the segmentation of myocardial abnormalities.

## 6. Code Availability

Updated versions of the anatomical network can be found at https://github.com/mic-dkfz/nnunet (accessed on 1 February 2021). The pathological network repository is available for download at https://github.com/KhawlaMarrakchi/Pathological-ICPIU-Net (accessed on 16 February 2022).

## Figures and Tables

**Figure 1 sensors-22-02084-f001:**
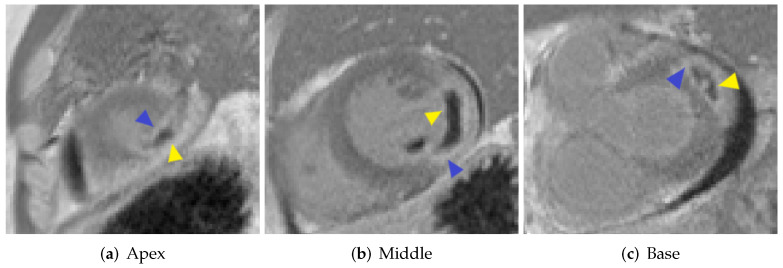
Short-axis LGE-MR images show MI (blue triangular) and MVO (yellow triangular).

**Figure 2 sensors-22-02084-f002:**
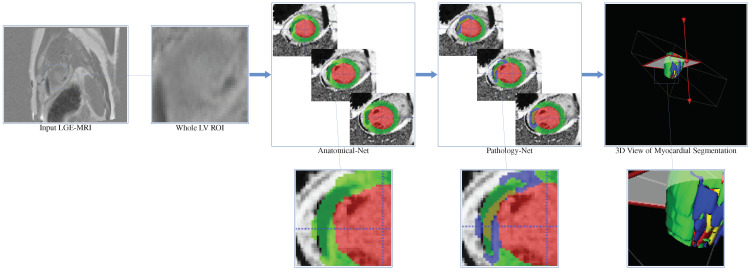
Workflow of our ICPIU-Net approach for fully automatic myocardial disease segmentation. The red, green, blue, and yellow colors show the LV cavity, the LV myocardium, scar, and MVO, respectively.

**Figure 3 sensors-22-02084-f003:**
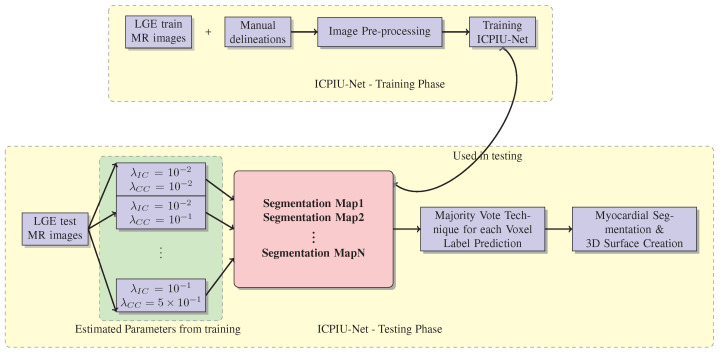
Block diagram of ICPIU-Net network.

**Figure 4 sensors-22-02084-f004:**
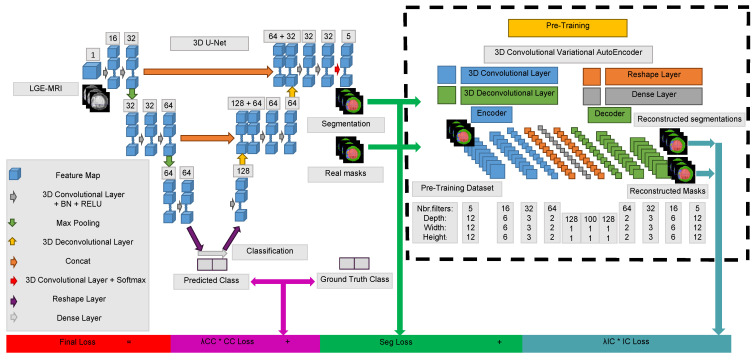
The overall architecture of our pathological segmentation network. We present the number of channels overhead each feature map. The classification is as well applied only in the training stage to supervise the segmentation network profoundly.

**Figure 5 sensors-22-02084-f005:**
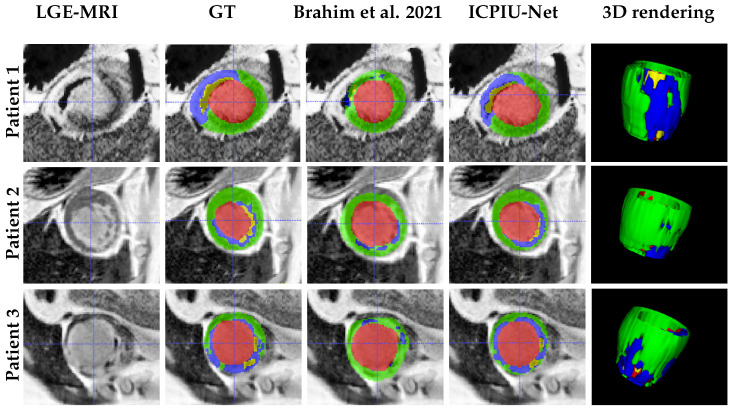
Qualitative results. In the first four columns, input LGE-MRI, manual delineations, and exemplary test results of the segmented myocardial regions in three slices extracted from LGE-MRI of three patients generated by the Brahim et al. [61] method and our ICPIU-Net approach are shown. The fifth column illustrates the 3D view of the myocardial tissues of our proposed method prediction. LV cavity is displayed in red, LV myocardium is in green, infarct in blue, and MVO in yellow.

**Figure 6 sensors-22-02084-f006:**
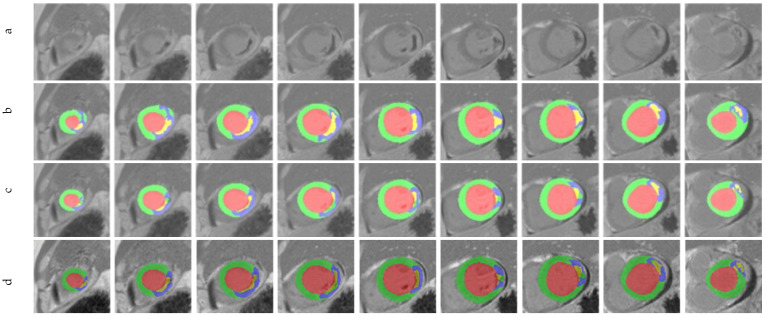
Segmentation results and the ground truth mask on Case 119. (**a**) LGE-MRI, (**b**) Ground Truth, (**c**), Zhang [47], and (**d**) ICPIU-Net.

**Figure 7 sensors-22-02084-f007:**
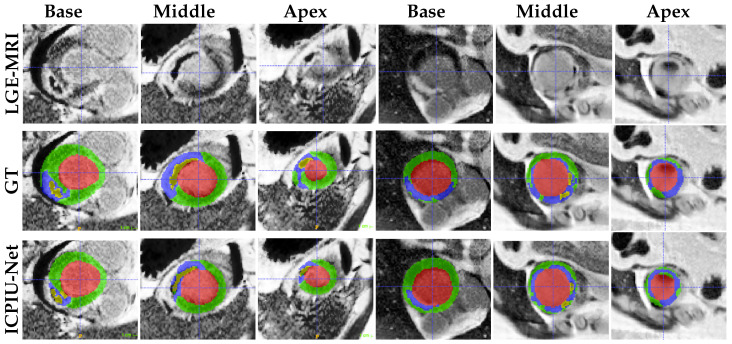
Examples of test segmentation results and ground-truth for three different levels (base, middle, and apex) of two patient slices (columns 1–3 from patient 1 and columns 4–6 from patient 2). Red: LV cavity, Green: LV myocardium, Blue: Infarct, and Yellow: MVO.

**Figure 8 sensors-22-02084-f008:**
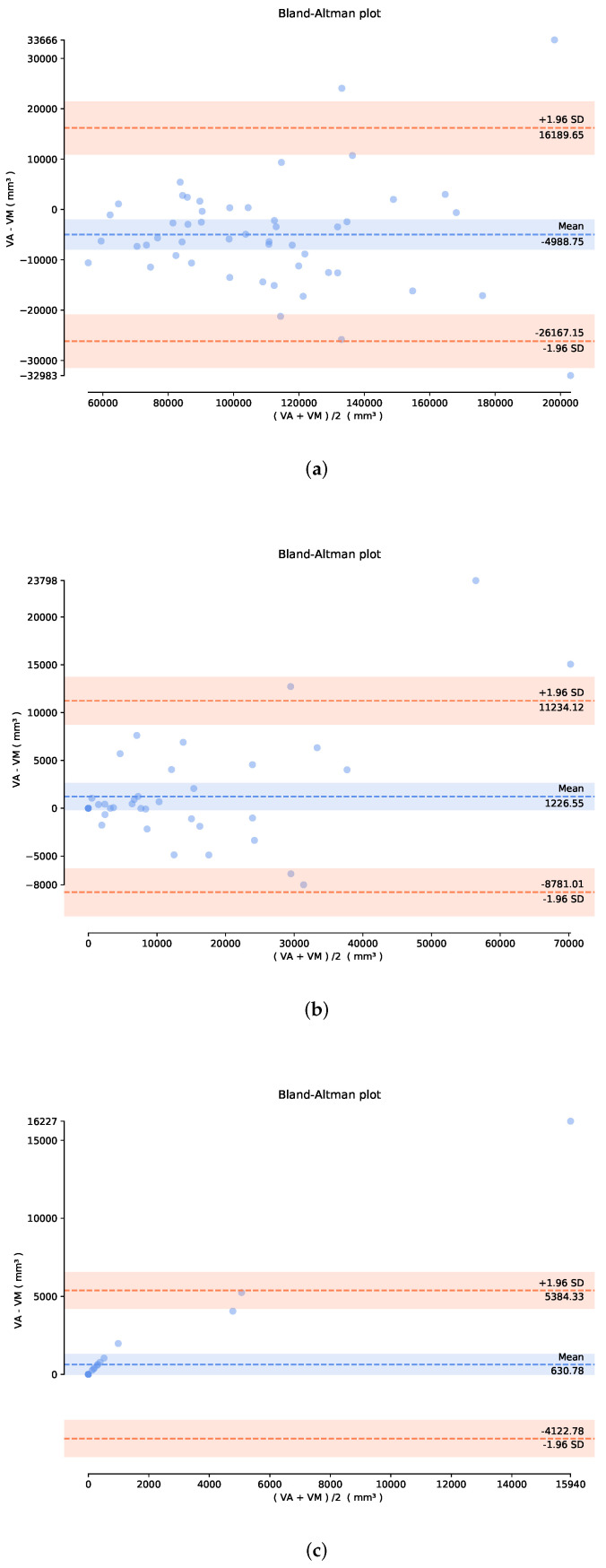
The graph represents the difference between the generated method and the expert target volumes according to their average. (**a**) Bland–Altman plot of LV myocardium volume acquired from our ICPIU-Net approach. (**b**) Bland–Altman plot of infarct volume acquired from our ICPIU-Net approach. (**c**) Bland–Altman plot of MVO volume acquired from our ICPIU-Net approach.

**Table 1 sensors-22-02084-t001:** Stratification of the EMIDEC dataset.

EMIDEC Dataset (*n* = 150)	Healthy Cases	Pathological Cases	
		Infarcted Cases	Infarcted + MVO (a Subclass of MI) Cases
Training dataset (*n* = 100)	33	27	40
Testing dataset (*n* = 50)	17	22	11

**Table 2 sensors-22-02084-t002:** Internal quantitative assessment on 5-fold cross-validation results.

Targets	Metrics	5-Fold Cross-Validation						
		Fold 0	Fold 1	Fold 2	Fold 3	Fold 4	Average	Standard Deviation
Myocardium	*DSC* (%)	95.38	95.07	95.21	95.35	95.59	95.32	0.17
	*AVD* (mm${}^{3}$)	232.74	290.61	225.42	229.14	203.49	236.28	29.01
	*HD* (mm)	4.02	4.78	3.87	3.61	3.46	3.95	0.44
MI	*DSC* (%)	77.05	79.45	78.73	78.92	77.34	78.30	0.75
	*AVD* (mm${}^{3}$)	283.31	267.26	190.34	156.53	271.25	233.74	50.65
	*AVDR* (%)	4.01	4.20	3.18	2.03	4.53	3.39	0.77
MVO	*DSC* (%)	76.54	79.15	79.92	75.51	78.03	77.83	1.62
	*AVD* (mm${}^{3}$)	34.18	26.80	45.10	49.19	46.68	40.39	8.51
	*AVDR* (%)	0.62	0.61	0.69	0.74	0.76	0.68	0.10

**Table 3 sensors-22-02084-t003:** A comparison of evaluation methods on 5-fold cross-validation of EMIDEC dataset. Best values are marked in bold font.

Targets	Metrics	Methods		
		Huellebrand et al. [44]	Zhang [47]	Proposed (ICPIU-Net)
Myocardium	*DSC* (%)	81.00	94.40	**95.32**
	*AVD* (mm${}^{3}$)	13655.55	6474.38	**236.28**
	*HD* (mm)	16.72	17.21	**3.95**
MI	*DSC* (%)	36.08	72.08	**78.30**
	*AVD* (mm${}^{3}$)	8980.5	4179.5	**233.74**
	*AVDR* (%)	7.07	3.41	**3.39**
MVO	*DSC* (%)	54.15	71.01	**77.83**
	*AVD* (mm${}^{3}$)	1501.73	918.69	**40.39**
	*AVDR* (%)	1.08	0.69	**0.68**

**Table 4 sensors-22-02084-t004:** Comparative study for EMIDEC myocardial segmentation in LGE-MRI (test leaderboard) [62]. Best values are marked in bold font.

Methods	Structures								
	Myocardium			MI			MVO		
	*DSC* (%)	*AVD* (mm${}^{3}$)	*HD* (mm)	*DSC* (%)	*AVD* (mm${}^{3}$)	*AVDR* (%)	*DSC* (%)	*AVD* (mm${}^{3}$)	*AVDR* (%)
Feng et al. [42]	83.56	15,187.48	33.77	54.68	3970.73	2.89	72.22	883.42	0.53
Huellebrand et al. [44]	84.08	10,874.47	18.3	37.87	6166.01	4.93	52.25	953.47	0.64
Yang et al. [46]	85.53	16,539.52	13.23	62.79	5343.69	4.37	60.99	1851.52	1.69
Zhang [47]	**87.86**	9258.24	**13.01**	71.24	3117.88	2.38	78.51	634.69	0.38
Camarasa et al. [41]	75.74	17,108.13	25.44	30.79	4868.56	3.64	60.52	867.86	0.52
Zhou et al. [48]	82.46	13,292.68	83.42	37.77	6104.99	4.71	51.98	879.99	0.54
Girum et al. [43]	80.26	11,807.68	51.48	34.00	11,521.71	8.58	78.00	891.13	0.51
Proposed (ICPIU-Net)	87.65	**8863.41**	13.10	**73.36**	**2693.84**	**1.95**	**81.31**	**511.25**	**0.32**

**Table 5 sensors-22-02084-t005:** Performance analysis and comparison between our proposed ICPIU-NET network without and with using *IC* and *CC* modules. Best values are marked in bold font.

Methods	Structures								
	Myocardium			MI			MVO		
	*DSC* (%)	*AVD* (mm${}^{3}$)	*HD* (mm)	*DSC* (%)	*AVD* (mm${}^{3}$)	*AVDR* (%)	*DSC* (%)	*AVD* (mm${}^{3}$)	*AVDR* (%)
Without *IC* and *CC*	**87.77**	9381.77	**13.07**	65.05	3096.54	2.39	78.82	553.56	0.34
Without *CC*	87.74	9201.04	13.09	71.71	2830.32	2.15	80.99	538.60	0.34
Proposed (ICPIU-Net)	87.65	**8863.41**	13.10	**73.36**	**2693.84**	**1.95**	**81.31**	**511.25**	**0.32**

**Table 6 sensors-22-02084-t006:** Additional metrics for EMIDEC myocardial segmentation [62]. Best values are marked in bold font.

Methods	MVO	
	Acc. (Case,%)	Acc. (Slice,%)
Feng et al. [42]	80.00	90.78
Huellebrand et al. [44]	70.00	85.75
Yang et al. [46]	76.00	81.56
Zhang [47]	**84.00**	**94.97**
Camarasa et al. [41]	74.00	84.36
Zhou et al. [48]	64.00	86.87
Girum et al. [43]	78.00	89.66
Proposed (ICPIU-Net)	**84.00**	**94.97**

**Table 7 sensors-22-02084-t007:** Results of the classification contest. Best results in bold.

Methods	Sensitivity (%)	Specificity (%)	Precision (%)	Accuracy (%)
Lourenço et al. [63]	87.88	70.59	85.29	82
Ivantsits et al. [64]	72.73	82.35	88.89	76
Sharma et al. [65]	72.73	41.18	70.59	62
Girum et al. [43]	78.79	88.24	92.86	82
Shi et al. [66]	90.91	94.12	96.77	92
Proposed (ICPIU-Net)	**100**	**94.44**	**96.96**	**98**

## Data Availability

Data available in a publicly accessible repository: the EMIDEC segmentation archive. Publicly available datasets were analyzed in this study. These data can be found at http://emidec.com/dataset (accessed on 1 April 2020).

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
