# Peer review of "An Improved 3D Deep Learning-Based Segmentation of Left Ventricular Myocardial Diseases from Delayed-Enhancement MRI with Inclusion and Classification Prior Information U-Net (ICPIU-Net)"

_sensors, 2022, doi:10.3390/s22062084_

Round 1
Reviewer 1 Report
+This manuscript is well written and easy to follow. The proposed method is sound.
+Even though the proposed framework is specifically designed for a dataset used in a challenge, its practical feasibility is overall good.
-The entire reference part is totally missing. Without that, it is difficult to properly judge the originality of this manuscript.
-The introduction length is not appropriate for a journal version. The authors are expected to provide a more detailed description of background and motivation.
-Table 3 is not very clear. What exact values are λ_{IC} and λ_{CC}?
-Table 4 shows that Zhang's results are very competitive. It would be interesting to see the visual comparison and a corresponding analysis between the proposed method and Zhang's method.
Author Response
Thank you for allowing us to submit a revised version of the manuscript “An Improved 3D Deep Learning-Based Segmentation of Left Ventricular Myocardial Diseases from Delayed Enhancement MRI with Inclusion and Classification Priors Information U-Net (ICPIU-Net)” for publication in the Sensors Journal. We appreciate the time and effort you dedicated to providing feedback on our manuscript and are grateful for the insightful comments and valuable improvements to our paper.
Please see the attachment.

Reviewer 2 Report
In this study, the authors proposed a deep learning framework for segmenting Left Ventricular Myocardial Diseases from Delayed-Enhancement MRI. The idea is to modify U-NET architecture to find an efficient model for this segmentation purpose. Although the idea looks ok, some major points need to be addressed as follows:
1. A critical concern is the use of a small sample size and this amount can not be enough to consider a good/unbias model.
2. The data was acquired with 1.5 and 3T Siemens MRI scanners, how did the authors perform standardization between two kinds of data?
3. How did the authors select the optimal hyperparameters of models (hyperparameter tuning process)?
4. Statistical tests should be conducted when comparing the performance results among models to show significant differences.
5. More discussions should be added in terms of clinical impacts of the model.
6. Source codes should be provided for replicating the study/method.
7. All references are displayed incorrectly. Please check again.
8. Deep learning has been used in previous biomedical studies i.e., PMID: 31920706, PMID: 31380767. Thus, the authors are suggested to refer to more works in this description to attract a broader readership.
9. Quality of figures should be improved.
10. Uncertainties of models should be reported.
11. EMIDEC challenge includes a classification problem also, why did the authors not try to conduct the classification problem after segmentation to see how it affects the performance?
Author Response
Thank you for allowing us to submit a revised version of the manuscript “An Improved 3D Deep Learning-Based Segmentation of Left Ventricular Myocardial Diseases from Delayed Enhancement MRI with Inclusion and Classification Priors Information U-Net (ICPIU-Net)” for publication in the Sensors Journal. We appreciate the time and effort that you dedicated to providing feedback on our manuscript and are grateful for the insightful comments and valuable improvements to our paper.
Please see the attachment.

Round 2
Reviewer 1 Report
Thanks for the authors' feedback. All my concerns are well addressed.
Author Response
Dear Mr. Rowan Teng,
Thank you for allowing us to submit a revised draft of the manuscript titled "An Improved 3D Deep Learning-Based Segmentation of Left Ventricular Myocardial Diseases from Delayed-Enhancement MRI with Inclusion and Classification Priors Information U-Net (ICPIU-Net)" to Sensors journal. We appreciate the time and effort you and the reviewers have dedicated to providing your valuable feedback on this manuscript. We are grateful to the reviewers for their insightful comments on our paper. We have been able to incorporate changes to reflect most of the suggestions provided by the reviewers. We have highlighted the changes within the manuscript.

Reviewer 2 Report
Thanks for addressing my previous comments. However, in my opinion, some comments have not yet been addressed well such as:
- A critical concern is the use of small sample size, but the authors did not show any solution for this part. I'm concerned that the model must contain some biases or overfitting. Thus, if the authors could not increase the sample size, some external validation tests should be tried.
- Hyperparameter tuning should be explained in more detail (i.e., evaluation method, hyperparameter range, best fit, ...).
- There was not any significant difference among models, how can the authors select the best model?
- Source codes should be provided for replicating the method/study.
- Uncertainties of model should be reported.
- EMIDEC challenge includes a classification problem also, why did the authors not try to conduct the classification problem after segmentation to see how it affects the performance? ==> I think this experiment should be conducted to have an additional comparison.
Author Response
Dear Mr. Rowan Teng,
Thank you for allowing us to submit a revised draft of the manuscript titled "An Improved 3D Deep Learning-Based Segmentation of Left Ventricular Myocardial Diseases from Delayed-Enhancement MRI with Inclusion and Classification Priors Information U-Net (ICPIU-Net)" to Sensors journal. We appreciate the time and effort you and the reviewers have dedicated to providing your valuable feedback on this manuscript. We are grateful to the reviewers for their insightful comments on our paper. We have been able to incorporate changes to reflect most of the suggestions provided by the reviewers. We have highlighted the changes within the manuscript.
Please see the attachment.

Round 3
Reviewer 2 Report
My previous comments have been addressed.